# Visualising the Dhammakāya through a Buddha Image: The Dhammakāya Text and Its Significance for Traditional Tai–Khmer Buddhist Practices

## Woramat Malasart

Religion Programme, University of Otago, Dunedin 9054, New Zealand; woramatediri072@gmail.com

**Abstract:** The *Dhammakāya Gāthā* is a Pāli Buddhist prose text that has been circulated within the cross-cultural/translocal sphere of Tai–Khmer Buddhism for over five centuries. Its earliest extant version appears on the "Braḥ Dhammakāya inscription", an engraved stone slab from the Stūpa of Wat Suea, Phitsanulok, Thailand, dated to 1549 CE. The *Dhammakāya* text consists of three parts. The first part identifies the knowledge and qualities/virtues of the Buddha with physical attributes of his body. The second part is the verses in praise of the Buddha's resplendent body qua the *dhammakāya*. The third section exhorts one in the *yogāvacara* lineage (a practitioner of spiritual discipline, i.e., a meditator) to recollect the *dhammakāya*, in order to attain the state of Buddhahood. The *Gāthā* was well known in the Tai–Khmer cultural sphere during the pre-modern period, but today, it is little used in modern practices. In this paper, I will analyse textual and paratextual elements of the *Dhammakāya Gāthā* to uncover the doctrinal meanings underlying the *Gāthā* and reveal the unique and unusual meditation practice called the *Dhammakāyānussati*, "Recollection of the Dhammakāya". I argue that the study of the *Dhammakāya Gāthā* enables us to understand the unique Buddhist practice: reciting [the *Dhammakāya* text], constructing [the image of the Buddha] and visualising [the *dhammakāya* embodied in the image], contributing to what we call "Buddhānussati" in the context of Tai–Khmer Buddhism.

**Keywords:** *Dhammakāya*; *Buddhānussati*; Buddha image; Tai–Khmer Buddhism

## 1. Introduction

Upon hearing the term "Dhammakāya" in the context of contemporary Thailand, most people would think of the "Dhammakāya" Temple located in Pathum Thani or/and Dhammakāya Meditation (Vijjā Dhammakāya) taught by Phra Mongkonthepmuni (1884–1959). However, in this paper, the term is used differently and reflects its actual use by Buddhists in the Tai–Khmer cultural sphere.[1] Many versions of the *Dhammakāya Gāthā* have been repeatedly copied, commented on, annotated and used by different groups of people over centuries within the cross-cultural sphere of Tai–Khmer Buddhism. The *Dhammakāya Gāthā*, the detail of which will be explained below, can belong to one or more genres of texts such as commentary, ritual, meditation and prayer, depending on the decisions of the users. The *Gāthā* can function to fulfil *multiple* parts of the core practices (meditation, consecration rites for Buddha images and Stūpas, commentarial exegesis and protective chanting) of Theravāda adherents within the cross-cultural sphere of Tai–Khmer Buddhism. However, this paper will focus on the relationship between the *Dhammakāya Gāthā* and a meditation practice, namely *Buddhānussati*.

The practice of *Buddhānussati*, "Recollection of the Buddha", is well known and common in Buddhist traditions. The detail of this mind training is described throughout Pāli literature, bilingual and vernacular texts. The way it is practiced varies from place to place, but it can be categorised into two main frameworks: recollecting the nine Buddha's qualities (as in *Visuddhimagga* and *Vimuttimagga* (Pitateeradhamm 2018, pp. 204–8)) and

visualising the body and/or image of the Buddha (described in the *Khuddakanikāya* of the *Suttanipāta* (Sn 1142) (Pitateeradhamm 2018, pp. 209–33)). Practising this kind of meditation will lead meditation practitioners to encounter the Buddha in the meditative state and finally to reach the awakening (Harrison 1978, pp. 36–40; Rotman 2009, pp. 177–96; Foxeus 2016, pp. 423–30; Pitateeradhamm 2018, pp. 209–51; Greene 2021, pp. 154–60).

The visualisation of the physical body (*Rūpakāya*) and/or an image of the Buddha is common in all Buddhist traditions, but that of the Dhamma Body (*Dhammakāya*) is unusual and less known in Buddhist practices, particularly in so-called "Theravāda Societies". In this paper, I will analyse the textual and paratextual elements of the *Dhammakāya Gāthā*—a rare and neglected Buddhist text—in order to uncover the doctrinal meanings underlying the *Gāthā* and identify the functional usages of this particular text related to the meditation practices. I will also seek to demonstrate how Buddhists in the cultural sphere of Tai–Khmer Buddhism construct the unconstructable image, in response to the textual call in the final part of the *Dhammakāya Gāthā* related to the recollection of the *dhammakāya*. I argue that examining the single family of Buddhist texts containing the "Dhammakāya Gāthā" enables us to understand a broad spectrum of Buddhist practices contributing to the practice of *Buddhānussati* or more precisely *Dhammakāyānussati* within the cross-cultural sphere of Tai–Khmer Buddhism.

## 2. Literature Review on the Study of the *Dhammakāya* Text(s)

The first scholarly study of the *Dhammakāya* text was published by George Coedès in 1956. Coedès transliterated a palm-leaf manuscript titled *Dhammakāya/Dhammakāyassa atthavaṇṇanā* (or DA), 'Body of the Dhamma/Explanation of the Meaning of the Body of the Dhamma.' The manuscript was from the Vajirañāṇa National Library of Siam. He transliterated it into romanised Pāli, and then translated it into French. He compared this manuscript with another from Vat Uṇṇālom, Phnom Penh (Cambodia), and identified only minor orthographic differences between the two. In his article, Coedès also mentioned another related Siamese manuscript, the *Suttajātakanidānānisaṃsa* (or SJNA), also from the Vajirañāṇa Library, and noted that a version of the *Dhammakāyassa atthavaṇṇanā* was contained in the second half of the thirteenth *phuk* or "bundle" (Coedès 1956, p. 258). In this article, Coedès did not pay much attention to the historical background of DA but did mention that the two copies of SJNA were of Siamese origin (Coedès 1956, p. 258).[2]

In 1961, Cham Thongkhamwan studied the *Braḥ Dharmakāya* inscription[3] found in the Stūpa of Wat Suea from Phitsanulok and dated ca. 2092 BE (1549 AD) (Thongkhamwan 1961, pp. 54–58). The inscription, written in Pāli with Khom-Sukhothai script,[4] was damaged, and today, only nine lines of text are legible. The text contains the core Pāli prose of the *Dhammakāya*. It describes the characteristics of the Buddha's body, which is made from *dhamma*s, "Truths", and is adorned with *dhamma*s. Thongkhamwan transliterated the inscription into modern Thai script and translated it into modern Thai. In his article, Thongkhamwan did not cite Coedès, but did refer to a manuscript from the Vajirañāṇa National Library, i.e., the manuscript in which Coedès called DA. Thongkhamwan called that manuscript (or a similar manuscript from the same collection) *Braḥ Dhammakāyādi* (or BD) and used it as a comparative source for his translation. Thongkhamwan did not discuss the historical background of the inscription or its ritual usage, but to date, this inscription is the earliest datable version of the *Dhammakāya* text.

In his 1992 work *Le chemin de Lanka*, Bizot discussed three *Dhammakāya* texts from Cambodia. These manuscripts he identifies as TK 217 from Vat Uṇṇālom in Phnom Penh; TK27 from Vat Chong Thnol in Phnom Penh; and TK 305 belonging to Achar Din in Phnom Penh (Bizot 1992, pp. 294–95).[5] He transliterated and translated one of the three *Dhammakāya* texts into French. Bizot argued that his manuscripts were similar to the Central Thai *Dhammakāya* text published by Coedès in 1956 and belonged to what he called the Yogāvacara tradition (Bizot 1992, p. 293).

In his 2004 book, *Becoming the Buddha*, Swearer refers to another version of the *Dhammakāya* text in northern Thailand located in *Tamra Kan Kosrang Phraphuttarup* (henceforth

TKKP), or "Manual for Making a Buddha Image" (Swearer 2004, pp. 50–73). Swearer compares this text with versions of the *Dhammakāya* text studied by Coedès and Bizot and identifies some differences between the three versions (Swearer 2004, p. 286). In Swearer's northern Thai text, the Buddha's *dhammakāya* has twenty-six characteristics. However, Coedès' text lists thirty characteristics and Bizot's text lists twenty-seven characteristics. Despite these and other differences, Swearer concluded that all three manuscripts were based on a single root text (Swearer 2004, p. 190).

In his 2013 PhD dissertation, Urkasame transliterated and translated a vernacular version of the *Dhammakāya* text into Thai and English (Urkasame 2013, pp. A366–80). This text was found in an undated palm-leaf manuscript titled *Gāthā Thammakāy*, 'Verses of the Body of Dhamma,' found in northern Thailand at Wat Pāsak Noi, San Kampaeng District, Chiang Mai Province. The manuscript was written using Tham Lanna script and composed of two parts: a Pāli section and its corresponding commentary in *Yuon* (Lanna). Urkasame indicates that there are similarities between the Pāli prose of this text, the sixteenth century SBD from Phitsanulok and the eighteenth century Golden *Braḥ Dhammakāya* inscription dated to the reign of King Rama I (Urkasame 2013, p. 14).

In the same year, Phrakru Palad Nayokworawat published an article titled "Dhammakāya in Braḥ Dhammakāyādi Scripture" (Nayokworawat 2013). He looked at a royal edition of the *Dhammakāya* text called *Braḥ Dhammakāyādi* (henceforth BD3). The *Braḥ Dhammakāyādi* was part of *Thepchumnum Tipiṭaka* produced during the reign of Rama III (1824–1851). In the article, Nayokworawat transliterated and translated BD3 from Pāli-Khom script into modern Thai. He pointed out similarities between BD3, Coedès' DA, Thongkhamwan's SBD, Bizot's *Dhammakāya* texts and Urkasame's GT (Nayokworawat 2013, pp. 8–10).

The most recent work that discusses the *Dhammakāya* text is Trent Walker's 2018 PhD thesis on Cambodian chanted leporellos (Walker 2018, pp. 113, 349–51, 417, 598, 789–91). Walker translated and analysed different versions of these texts in Cambodia, which are often called *Gāthā Pañcuḥ Braḥ Lakkhaṇa*, "the verse of the incantations for implanting the sacred marks" (Walker 2018, p. 131). In his thesis, Walker did not list the *Dhammakāya* texts in Thailand, but did list the indexes and diplomatic editions of the *Dhammakāya* texts in Cambodia (Walker 2018, pp. 1017–19). In terms of *kammaṭṭhāna* practice, Walker commented that

> The closing lines of this text [the *Dhammakāya*] make clear that the desired soteriological aim is to become the Buddha oneself . . . In this case, the implication is that certain *kammaṭṭhāna* meditation practice can lead directly to Buddhahood (Walker 2018, p. 598).

According to the existing scholarship, some editions of the *Dhammakāya Gāthā* can be dated and/or located geographically. Undated documents are also useful for understanding the range of content and historical background of the *Dhammakāya* text. A number of *Dhammakāya* manuscripts that have been discovered today are dated to the pre-modern Siam and this reflects their popularity at that time. However, today, the *Dhammakāya* texts and their associated rituals have received little attention from the modern practitioners. Moreover, some do not even realise that the *Dhammakāya Gāthā* has previously existed in the cross-cultural sphere of Tai–Khmer Buddhism. In the next section, I shall give a brief overview of the *Dhammakāya Gāthā* and analyse its textual elements in order to reveal doctrinal meaning underlying this text.

## 3. The *Dhammakāya Gāthā*

The documents that record the texts of the *Dhammakāya Gāthā* have been found in central Thailand, southern Thailand, northeastern Thailand (Isan), northern Thailand (Lanna) and Cambodia. The earliest datable document, in which the *Dhammakāya Gāthā* appears, is the "Braḥ Dhammakāya inscription", an engraved stone slab mentioned above from the Stūpa of Wat Suea, Phitsanulok, Thailand, dated to 1549 CE (Thongkhamwan 1961, pp. 54–58). The *Dhammakāya Gāthā*, recorded in Khom, Tham, Mūl and Thai scripts, has three parts. The first

part ("personification") appears in the Thai recension of the *Manorathapūranī*. The first part compares *ñāṇa*, the "knowledge", and *guṇa*, "qualities/virtues", of the *dhammakāya* with the physical attributes of the Buddha's body (see Figure 1). Twenty-two or twenty-six of them (depending on the versions) are identified with the Buddha's physical attributes (beginning with the head and ending with the feet) while the other four elements are identified with his robe. Some identifications contain doctrinal significance and a numerical correspondence (e.g., the equations between the teeth and thirty-seven factors of awakening, between the (four) eye teeth and the Knowledge of Four Noble Paths and between the (ten) fingers and the Knowledge of Ten Recollections). Other parallels are drawn between anatomical characteristics and supernatural organs (e.g., the equations between the eyes and the (five) supernatural eyes, and between the ear and the divine ear) (Reynolds 1977, pp. 385–86). In minor cases, the identifications are made based upon the combination of anatomical appropriateness and metonymy or verbal congruence, for instance, the equations between the feet (*pāda)* and Four Paths of Accomplishment, *caturiddhipādañāṇa*. Here, *pāda* could also be rendered as the "path" or "basis".

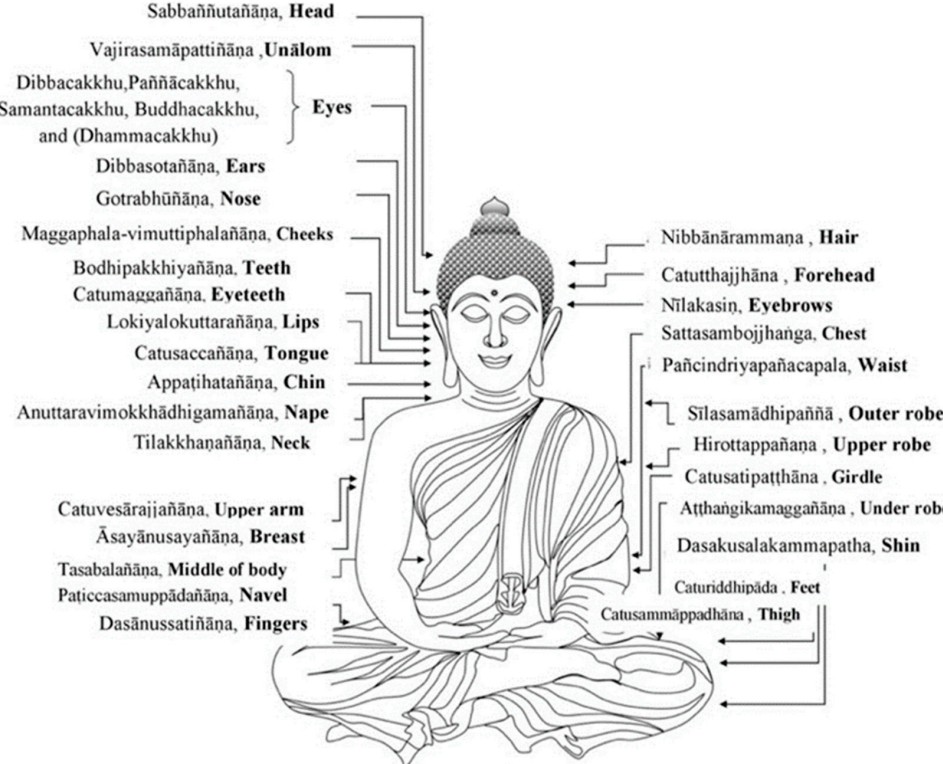

**Figure 1.** Each element of the *dhammakāya* described in the personification part of the *Dhammakāya Gāthā* is compared to the Buddha's attributes (Malasart 2021, p. 83).

The second part ("glorification") is followed by verses in praise of his resplendent body *qua* the *dhammakāya*. Literally, this part suggests that the Buddha outshines all other gods and humans (*aññesaṃ devamanussānaṃ buddho ativirocati)*. The third section ("summarising") concludes the descriptions in the first and second parts of the *Dhammakāya Gāthā* and instructs the practitioner of spiritual discipline (i.e., a meditator in the *yogāvacara* lineage) to recollect the *dhammakāya*. I consider this part the most important section in the text, and it shall be presented below. The below passage is taken from a palm-leaf manuscript titled *Ṭīkā Braḥ Dhammakāya* (Rong Srong Edition, TBD-1)[6] from the reign of Rama I (1782–1809), i.e., that of Phra Phutthayotfa Chulalok. The transliteration and the translation of the *Dhammakāya Gāthā* are presented in my MA thesis (Malasart 2019, pp. 49–63) and 2021 article (Malasart 2021, pp. 81–82).

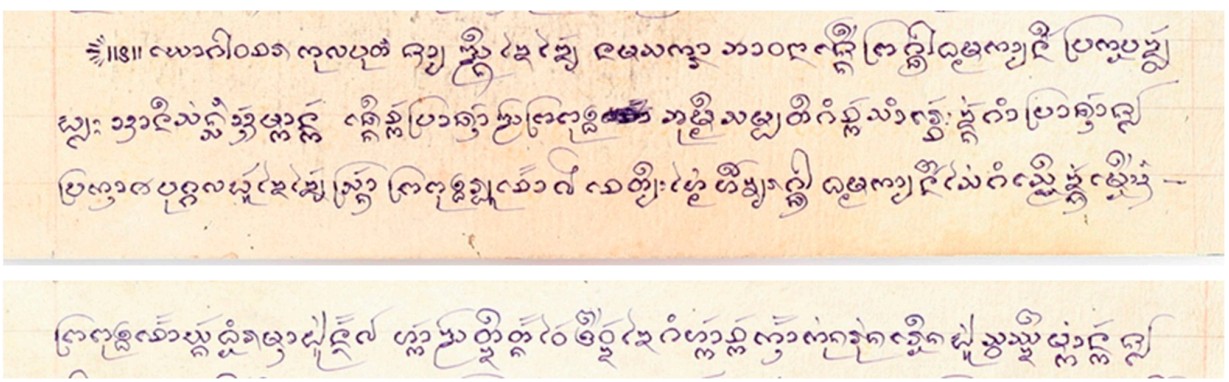

(Folio ka/v).

> [Diplomatic Transcription] . . . dhammakāyabuddhalakkhaṇaṃ yo⊙gāvacarakula puttenatikkhañāṇenasubbaññubuddhabhāvaṃpatthentena puānappunaṃanussa ritabbaṃ | | . . .

> [Standardised Edition] . . . dhammakāyabuddhalakkhaṇaṃ yo⊙gāvacarakulaput tena tikkhañāṇena subbaññubuddhabhāvaṃ patthentena pu⊙nappunaṃ anussa ritabbaṃ | | . . .

> [Translation] . . . This Buddha's characteristic is the *dhammakāya*. [This] should be remembered again and again by a noble man of a good family, practicing spiritual exercises, possessing sharp Knowledge and aspiring to become an omniscient buddha . . .

## 4. Construct the Unconstructable Image: Visual Representation of the *Dhammakāya*

"Dhammakāya" or "Dharmakāya" has been a problematic term in Buddhist Studies and Buddhology for decades. There is no single interpretation for the term, but the term "refers in any instance to a truer mode of the Buddha's existence than any impermanent, physical body of his that was seen and heard in the world (Jones 2020, p. 129)". Scholars, basing their interpretations on a textual analysis, translated this term as the "Body of Dhamma", the "Body of Teaching", the "Body that can be attained in deep meditation" (Reynolds 1977, p. 386), the "Body of Enlightenment" (Jantrasrisalai 2009) and the "Collection of Teaching (Collins 2014, p. 259)".

One might be curious as to how the *dhammakāya* is constructed, since it was understood by some scholars as a formless body, an invisible body and a collection of the Buddha teachings preserved in the *Tipiṭaka*. In this section, I shall present the way that the *dhammakāya* is embodied during a Buddha image construction in Lanna (see Figure 2), reflecting how the concept of the *dhammakāya* is understood by Buddhists in the region.

It is stated in the Lanna manuscripts (Siripunyo 2015, pp. 39–40) that before an image of the Buddha is constructed, one should first construct the *dhammakāya*, which is a noble body of the Buddha, or *varakāya* (vara, "noble/sublime" + kāya, "body"). A good example of the paratext that includes both official and individual usages of the *Dhammakāya* text is located in *A Manual for Installing a Buddha's Heart and Cetiya* (**MIBHC**) composed by northern Thai monk Kruba Kong (1902–1989).[7] He wrote the following:

A Manual for Installing a Buddha's Heart and Cetiya Written by Master Kong (folio 5).

… โยคาวจรกุลบุตรชายหญิงใดได้นมัสการ　ภาวนาเถิงพระคาถาธัมมกายนี้　ประกอบด้วยผลอานิสงส์อันมากนัก
เถิงจักปรารถนาเอาพระพุทธภูมิสัมบัติ ก็จักสำเร็จดั่งคำปรารถนา ชแลฯ ประการ ๑ ปุคคลผู้ใดได้สร้างพระพุทธรูปเจ้า
แลเจดีย์ใหม่ หื้อเขียนคาถาธัมมกายนี้ใส่ ก็เสมอดั่งเมื่อพระพุทธเจ้ายังธอรมาน (ธรมาน) อยู่นั้นแล หากเอาติดตั้งไว้
ที่วัดใดก็หากจัก ก้านกุ่ง รุ่งเรือง อยู่สวัสดี มากนักชะแล

. . . One in the lineage of *yogāvacara-s*, i.e., a male or female meditation practitioner—
who venerates, recites, or contemplates on the *Dhammakāya Gāthā*—will gain
great fruitful merit, living prosperity, and even attain the state of an omniscient
buddha, if they wish. For one instance, if one wishes to construct a new Buddha
image and a Cetiya, the *Dhammakāya Gāthā* should be installed therein. By doing
so, it will be like the time when the Buddha himself is still alive (dharamāna).
Moreover, wherever, such as a temple, this *Gāthā* is established, it will bring
prosperity and happiness.

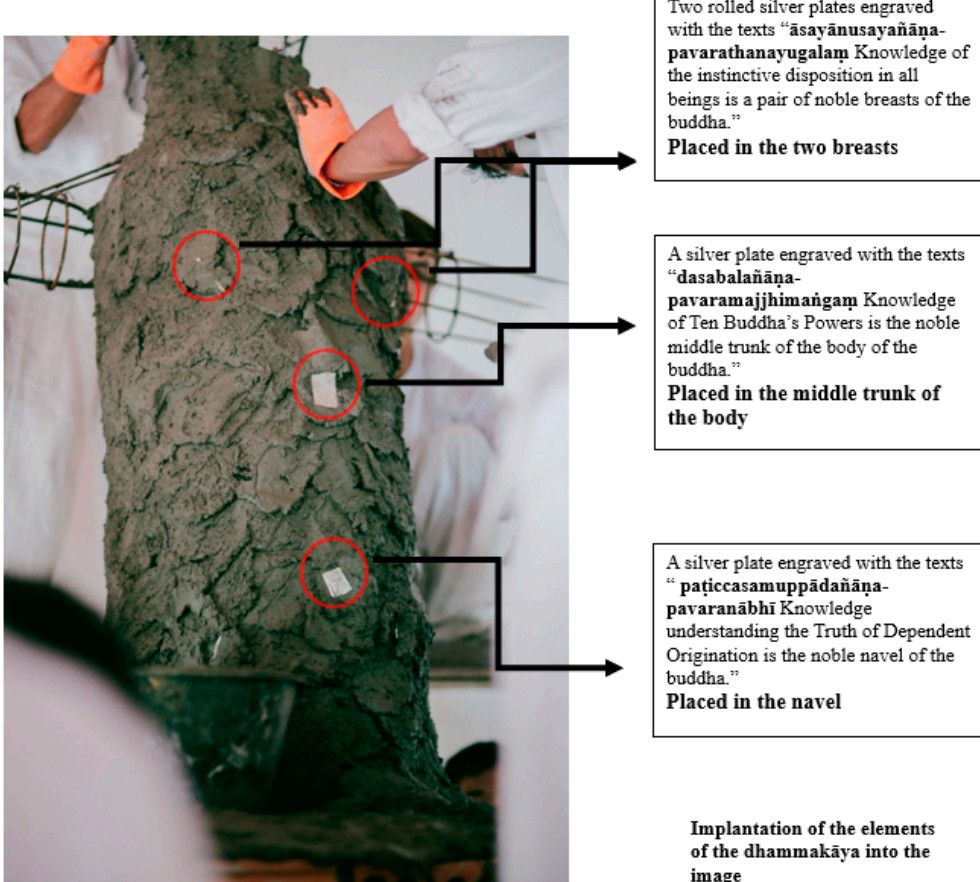

**Figure 2.** The implantation of the elements of the *dhammakāya* into the Buddha image (Malasart 2022,
p. 74). Note that each mark implanted in the image's part corresponds with the textual components
of the *dhammakāya* text in Section 3.

According to the above quotation, the first interesting point to note is that the term
"yogāvacara-kulaputta" here is not restricted to the *son* of a good family but includes
all spiritual practitioners, both men and women. Clearly, this above paratext suggests
two main functions of the *Dhammakāya Gāthā*, and the goals of these performances are
associated with two aspirations: mundane and supramundane attainments. As for the
mundane wish, the *Gāthā* is recited ("bhāvanā" in this sense means reciting selected words
within one's mind, not out loud) so that the practitioners would gain prosperity in life, and
alphabetical elements of the *Gāthā* are not just symbols to record Buddhist doctrine about
the *dhammakāya* but are used significantly to consecrate or empower a Buddha image and a

Cetiya. Practitioners believe that wherever the *Gāthā* is established, it will bring auspicious things and happiness to the place, such as a temple, and the people who are in the place. As for the supramundane purpose, which is consistent with that described in the text itself, the paratext suggests that the practitioners who wish to become an omniscient Buddha in the future should venerate and/or recite the *Dhammakāya Gāthā*. As a consequent fruition, their wishes will be fulfilled.

Another important point that needs to be addressed is the concept of "Dharamāna". This term can be rendered as "lasting", "continuing" and "living". Steven Collins proposes that the term "Dharamāna" (and *ṭhita*) is associated significantly with the term *Rūpakāya*, referring to the body of the Buddha while he is still alive (Collins 2014, p. 260). However, as presented in the above passage, the term "Dharamāna" is not used to refer to the *Rūpakāya* but the *Dhammakāya*. The connection between "Dharamāna" and "Dhammakāya" highlights that although the Buddha has passed away and gone to *Nibbāna*, and that his *Rūpakāya* is no longer present, his *Dhammakāya* is present, making him as if he is still alive (Dharamāna) in the world.

As related to the contemporary practice, according to Phra Pongsakon, an informant interviewed during my fieldwork undertaken in 2021 at Chiang Mai, in the past, the *Dhammakāya Gāthā* was the most important *Gāthā* used when making a Buddha image. Without this *Gāthā*, the image would be understood as just a lump of concrete. This has changed, and modern ritual experts no longer use the *Dhammakāya Gāthā* during image construction. Phra Pongsakon showed me photographs of the silver plates engraved with Pāli texts of the *Dhammakāya* that he used when constructing a main Buddha image in 2017 (see Figure 3). He explained that in addition to the *Dhammakāya Gāthā*, there were two other Pāli *Gāthā*-s, the *Asītayānubyañjana*, also known as *Anubyanñjana*, "eighty minor marks", and *Mahāpurisalakkhana*, "thirty marks of a great man", that should be inscribed onto plates. After the *Gāthā-s* are inscribed onto plates, they are rolled up and put in a golden casket. When the construction of the Buddha image is almost finished, the golden casket is inserted near the navel of the image (from the back). According to Phra Pongsakon, the golden casket symbolised the thirty elements of the "dhammakāya", and the texts or plates inscribed with the thirty-two *mahalakkhana*, "major marks", and eighty *lakkhana* or "minor signs" correspond to the *Gāthā*. It is unclear why the *dhammakāya* is placed around the navel of the image, but this performance re-enacts the scriptural claim in the Ayutthaya Meditation Manual (AMM). The historical background and details of the manual were analysed (Cholvijarn 2021, pp. 66–70). The AMM manuscript reads

> "… พระโยคาวจรผู้รู้ว่า ธรรมกายดำรงอยู่ในหทัยประเทศ แห่งสรรพภูตทำให้เหมือนดังว่าหุ่นยนต์ ท่านจึงตั้งใจเจริญพ ระวิปัสสนาญาณ เพื่อให้ถึงธรรมกาย เป็นที่พึ่งอันยอดเยี่ยมโดยสิ้นเชิง ถึงสถานอันสงบระงับประเสริฐเที่ยงแท้ เพราะควา มอำนวยของธรรมกาย นั้นเป็นอมตะ ฯ …"

> "... those in the lineage of *yocāvacara* realise that the *dhammakāya* is present in the heart sphere/centre/core (*hadaya-pradeśa*)/nevel[8] of all beings (*sabbabhūta*), causing them to move about like a machine-driven puppet. Therefore, the meditators intend to cultivate the insight to reach the *dhammakāya*, which is completely the most supreme refuge. It is calm, peaceful, sublime, and permanent because the nature of the *dhammakāya* is immortal ..." (Phramaha Jai Yasothornrat 1935, p. 283).

The above quotation suggests that the *dhammakāya* is attainable and could be visioned during the insight (Vipassanā) meditative state. Perhaps, after encountering the *dhammakāya* during the meditation, the author of the *Dhammakāya* text wrote that, regarding *dhammakāya-buddhalakkhaṇaṃ*, "this set of Buddha marks is [called] the dhammakāya", in order to instruct what the *dhammakāya* looks like. When this idea was transmitted in the ritual context, the set of the *dhammakāya*'s marks was installed into the marks of the Buddha image, demonstrating that, virtually, the *dhammakāya* is a Buddha likeness.

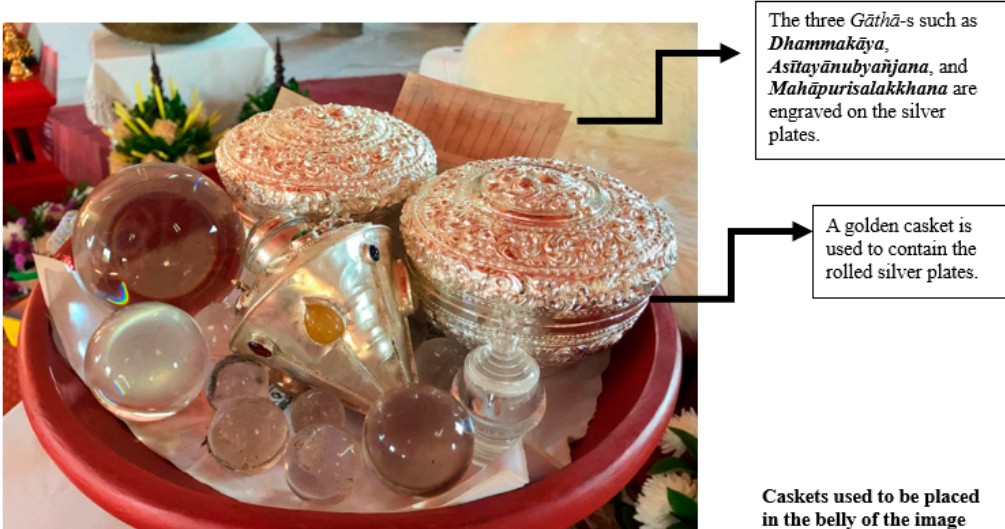

The three *Gāthā*-s such as *Dhammakāya*, *Asītayānubyañjana*, and *Mahāpurisalakkhana* are engraved on the silver plates.

A golden casket is used to contain the rolled silver plates.

Caskets used to be placed in the belly of the image

**Figure 3.** A golden casket containing the three *Gāthā*-s was placed in the belly of the image. Other valuable objects such as crystal balls and the heart of the Buddha made from silver and physical body relics were also placed inside the image (Malasart 2022, p. 73).

When examining the concept of the "dhammakāya" in the *Dhammakāya* text (see Section 3) and looking at how it is used during the Lanna Buddha image construction, this reflects that the image of the Buddha, considering it as the *dhamma*, "does not mean that the material form which impacts on the sense of sight to create a visual image is itself the Buddha, but rather that to be the image must be the *dhammakāya*" (Swearer 2004, p. 72). In other words, it can be said that the image is not just a material representation of the Buddha, but of the *dhammakāya*.

I also argue that the *dhammakāya* is not a collection of teachings preserved in Buddhist scripture, but the Body of Knowledge or a sublime body (varakāya) of the Buddha or a Buddha likeness body that consists of a head, hair, eyes, ears, legs, etc., and is significantly associated with meditation practices. The *dhammakāya* is not a physical form but was used as a mental object that keen *yogāvacara*-s, "meditation practitioners", could remember and use when practicing meditation. The process of recollecting the *dhammakāya* (here, embodied in the Buddha image) will lead the *yogāvacara* directly to attain (real) *dhammakāya* and become the omniscient Buddha (Walker 2018, p. 598). In other words, the *dhammakāya* transforms the practitioner to become a Buddha.

## 5. Recollecting the *Dhamamkāya* by Visualising a Buddha Image

As related to the meditation context, Coedès, Bizot, Crosby, Urkasame, Choompolpaisal, Walker and others have linked the *Dhammakāya* text to *yogāvacara* tradition or *Boran Kammaṭṭhāna*.[9] The reason why they link *yogāvacara* tradition to the *Dhammakāya* text is still unclear. However, one possibility is the existence of the term "yogāvacara" in the third part of the *Dhammakāya* text (. . . **yogāvacarakulaputtena** *tikkhañānena sabbaññūbuddhabhāvaṃ* . . .) and the application of *Abhidhamma* within the *Dhammakāya* text, corresponding with some key features of the *yogāvacara* tradition listed by Bizot and Crosby (the importance of *Abhidhamma* categories and the books of the *Abhidhamma Piṭaka*) (Crosby 2000, p. 141).

Clearly, the third part of the *Dhammakāya* text itself asserts a son from a good family who possesses the "sharp insight Knowledge" (not just an ordinary person) should visualise/recollect (*anussarita*) the *Dhammakāya* regularly, in order to attain the state of an omniscient Buddha. The recollection or visualisation of the *dhammakāya* is not common in the Pāli literature, but that of the Body of the Buddha or Buddha image can be found in both Pāli text and vernacular meditation manuals.[10] In the Lanna Buddha image construction, as I have shown in the previous section, the *dhammakāya* and the Buddha image are closely and significantly associated with one another.

The *dhammakāya*—understood as the body, in contrast with the physical body of the Buddha—is embodied in the Buddha image and makes it present both virtually and physically as a Dhamma puzzle, responding to the call in the final part of the *Dhammakāya* text. The *dhammakāya* should be used as a mental object to lead the practitioner to the state of an omniscient Buddha. Since the *Dhammakāya* is embodied in the image, it implies that "visualising the dhammakāya" is somehow "visualising the Buddha image" and that belongs to one type of *Buddhānussati*, "Recollection of the Buddha". The visualisation of the Buddha image, not merely the recollection of the Buddha's qualities, is another way to practice the *Buddhānussati* in Lanna.[11] This practice appears in the traditional Lanna meditation text named *Kammaṭṭhāna, Bhāvanā, Gāthā Thammakāy* (KBG) dated to 1895, where the *Dhammakāya Gāthā* is also located. The text reads

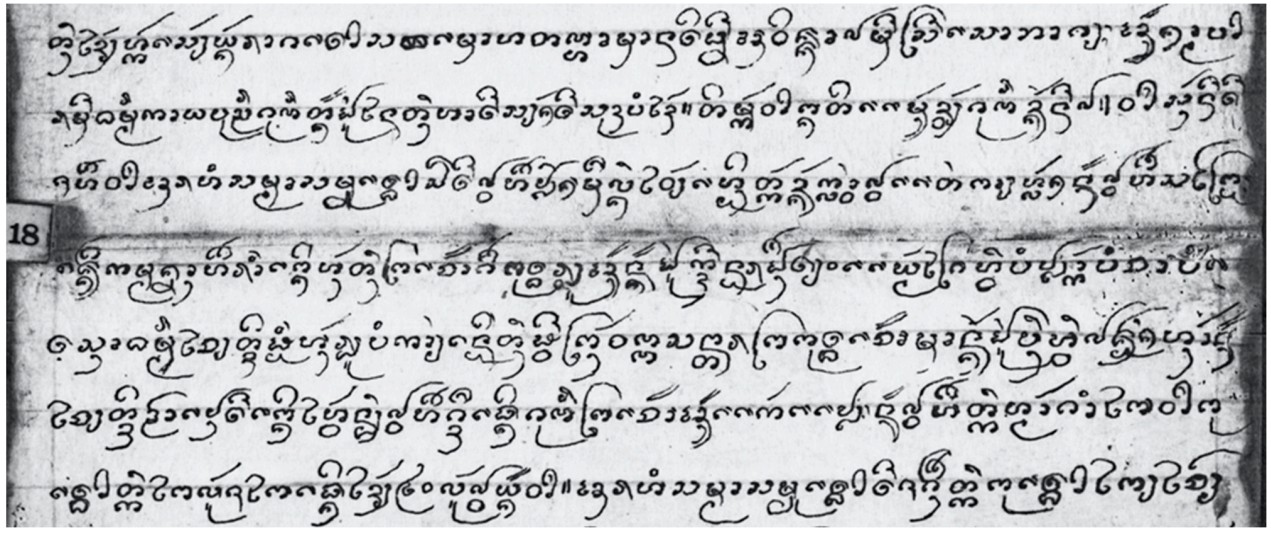

(Folio 18).

[Standardised Edition]

… [ภควา] ตนได้หักเสียยังราคะ โทสะ โมหะ ตัณหา มานะ ทิฏฐิ อวิชา แล มีศรีโสภาคย์ อันงาม[ผู้มี]บารมีธัมมกาย บุญคุณตั้งอยู่ในตน หาที่เสี้ยง ที่สุดบ่ได้ | ติ มักว่า อิติ แม้นด้วยคุณตั้งนี้แล |ว่าฉันนี้ที ๑ ให้ว่า อรหํ สมฺมาสมฺพุทฺโธ ๓ ที แล้วให้ปลงมือลงไว้เหนือตักดังกล่าวแล้วแต่ภายหลังนั้นแล้ว ให้เฉพาะซึ่งกัมมัฏฐาน ให้ร่ำเพิงเห็นตนพระเจ้าคือพุทธรูปอัน นั่งอยู่ก็ดีนอนยืนเทียว แย้มใคร่หัวปากบ่จาบ่เทศนาธรรม ใจติดผ่อ เห็นรูปบ่ก่าย เนื้อตนผิวพรรณวรรณะ สันถาระ พระพุทธเจ้ามานั่งอยู่บนหัวแลซ้องหน้านั้น ใจติดเอาเป็นที่พึ่ง …

[Translation]

… [*bhagavā*] who destroys *Rāga, Dosa, Moha, Taṇhā, Māna, Diṭṭhi*, and *Avijjhā*, who is possessed of luck, the perfection of "Dhammakāya" [which is] the set of quality (*Guṇa*) within himself, which is endless and infinite | *ti mok wa iti* with this quality | [the meditation practitioner] recites, at first, *arahaṃ sammāsambuddho* 3 times, then, rests [two] hands on the lap. After that, the practitioner starts performing the [following] particular *Kammaṭṭhāna*. He visualises the body of the Buddha [the Buddha image] choosing one preferred position, which could be sitting, sleeping, standing, or walking, with his smiling, but without talking or preaching the Dhamma. The mind of the practitioner absorbs the selected image [mentally and clearly see the image], seeing it in proper size and adorning with bright colour. [Once seeing the Buddha image clearly in the mind], the meditation practitioner locates the mental Buddha image to the top of his head and then places it in front of him. He lets his single-minded concentration attach to it without any disturbances.

The above translation is part of the Lanna meditation manual KBG. This part explains how the *Buddhānussati* is practised in the region. It starts with the recollection of Buddha's

*guṇa* (*iti pi so bhagavā arahaṃ sammāsambuddho . . .* (Pitateeradhamm 2018, pp. 204–6)). Then, the practitioner develops the mind into the visualisation of the Buddha image. When the practitioner clearly visions the image in the mind, they then locate the mental image on different parts of the body. This visualisation practice shares some similarities with traditional Burmese meditation, "I am the Buddha, the Buddha is me" (Foxeus 2016, pp. 438–41), and Chan meditation, "Bring to mind the Buddha", in China (Greene 2021, pp. 154–60).

## 6. Recollecting the *Dhammakāya* by Reciting the *Dhammakāya Gāthā*

As addressed in Kruba Kong's manual in Section 4, one way to recollect the *Dhammakāya* is to recite the textual element of the *Dhammakāya Gāthā*. In other words, to recollect the *dhammakāya*, one could recite the qualities of the *dhammakāya* recorded in the *Dhammakāya* text. This form of practice is consistent with the title of the chant in central Thailand, "Dhammakāyānussati-kathā" [Dhammakāya + anussati + kathā], or the Word of Recollection of the *Dhammakāya*. While reciting the textual elements of the *Dhammakāya Gāthā*, I speculate that those who understand the doctrinal meaning of *dhammakāya* depicted in the *Dhammakāya* text would be able to visualise or recall the image of the *dhammakāya* into their mind, creating the mental image of the *dhammakāya*.

As we can see in Kruba Kong's manual, the recommendation for the recitation of the *Dhammakāya* text is highlighted with the goals of gaining prosperity in life and achieving the state of Buddhahood. A similar practice can be found in central Thailand and Lan Xang. For example, in central Thailand, the *Dhammakāyānussati-kathā* (DK) located in a traditional chanting book (*Suat Mon Plae*) and *Namasakāra Braḥ Dharmakāya* (บทนมัสการพระธรรมกาย), part of the Buddhist Chanting Leporello (BCL) preserved at the Central Library Chulalongkorn University, is meant to be recited somehow in ceremonies—although the paratexts in the manuscripts do not suggest their functional uses associated with personal recitation. The reason for this claim is because they are located in the chanting books, and their titles are related to the performative context, for instance, *Dhammakāyānussati-kathā* means Word [to be recited] for Recollecting the *Dhammakāya* and *Namasakāra Braḥ Dharmakāya* means the Prose to [be recited to] Venerate the *Dhamamkāya*.

In Lan Xang, *A Manual for Installing Yantra into a Buddha Image* from Khon Kaen (MIYB) suggests the practitioners to recite *the Dhammakāya Gāthā*, although the main functional use of the *Gāthā* is closely related to the Buddha image construction. The text states the following:

| ปุญฺณปุญฺณํ | **puṇṇapuṇṇaṃ** |
|---|---|
| สวดทิยายขึ้นใจ นบอย่าได้ทดคี่จักได้เถิง ยังนีรพานเป็นที่แล้ว ตามหากได้กระทำมาแล | [The Yogāvacara] should recite [the *Dhammakāya Gāthā*] and remember it by heart on a daily basis. This, as its fruition, would support practitioners to attain the paths to Nibbāna in accordance with what they have done. |

It is significant to note here that this supramundane goal recorded in this Lan Xang manuscript is slightly different from that addressed in Kruba Kong's manual and that recommended by the text itself (to attain the Buddhahood). The goal of the Lan Xang manuscript is to attain Nibbāna in the future, which is not restricted to the state of "buddhahood", but "arahatship". This reflects the localisation idea (Wolters 1999) that the ways local practitioners use the same text can vary from location to location, and even sometimes beyond what the text itself instructs.

Similarly, although the bilingual Pāli-Lanna *Dhammakāya* text in the *Manual for Making a Buddha Image* from Wat Chiang Man, Phayao (MMBI) is meant to be used in the Buddha image construction according to the ritual instruction addressed in the paratext, in the last part of the text, the vernacular translation of the Pāli term *puṇṇapuṇṇaṃ* (*punappunaṃ*) "again and again, repeatedly, regularly" puts the text into another context of individual recitation.

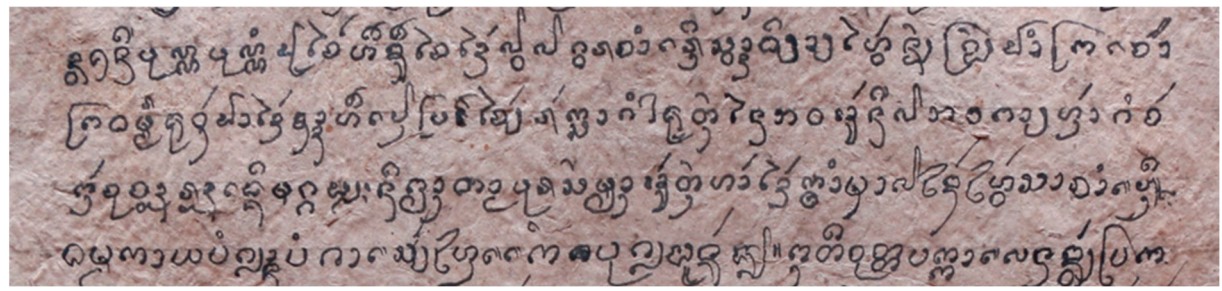

(Folio 9).
[Standardised Edition]

… ปุณฺณปุณฺณ อยู่ไจ้[ๆ] หื้อ[จำ]ขึ้นใจได้แล้วแล ควรจำเริญสวาธิยายไหว้นบครบอำพระเจ้า

พระธัมม์ชุวันอย่าได้ขาด หื้อเป็นปัจจัยรักษาค้ำชูตัวในภาวะอันนี้แลภาวะภายหน้าก็จัก

พลันจุจอดรอดเถิงมัคคผละ นิพพานตามบุญสมภารอันตนหากได้กระทำมาแล

ได้ไหว้สาจำเริญธัมมกายบ่คลาดบ่คลาเสียไหนแก่บุคคลผู้นั้นชะแล …

[Translation]

… *puṇṇapuṇṇaṃ* again and again, it should be remembered by heart, and then recited to venerate the Buddha and Dhamma without any missing days. This, as its fruition, would support practitioners this lifetime and to attain the paths to Nibbāna in the future depending on their past good deeds. It will be successful certainly for those who venerate and recite the *Dhammakāya Gāthā* …

This instruction is also shared in the content of *Namasakāra Dhammakāya* from Lumphun [KB]. In the first instance, after the Pāli *Dhammakāya* Core, the manuscript reads

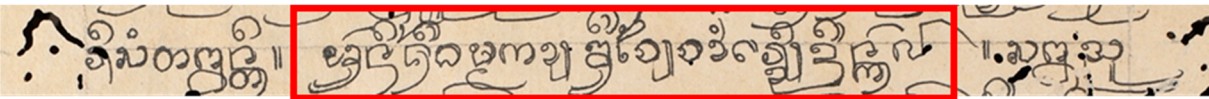

(Folio 98).
[Standardised Edition]

ริตฺตพุพนุติ// อันนี้ชื่อ ธัมมกายขึ้นใจจำเริญดีนักแล //สพฺพญฺญู

[English Translation]

This is called the "Dhammakāya [Gāthā]". It should be remembered by heart so as to gain prosperity.

The same as MMBI, for the second instance, the author of this bitext elaborates on the term *puṇṇapuṇṇaṃ* (*punappunaṃ*) with special and unique explanations. It suggests how the *Dhammakāya* text is used in a practical orientation.

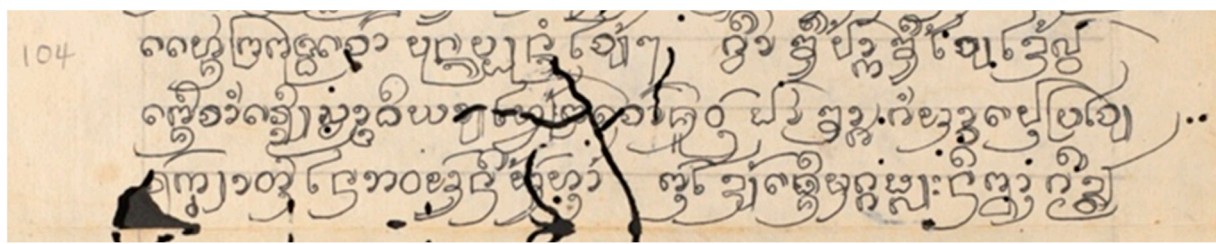

(Folio 104).
[Standardised Edition]

แห่งพระพุทธเจ้า ปุนุนปุปุนุนํ ไจ้ๆ คือว่าขึ้นปากขึ้นใจได้แล้ว

เพิงจำเริญสวาทธิยายไหว้พระเจ้าฮูวัน อย่าขาดก็อาจเป็นประจัย

รักษาตนในภาวะอันนี้อันน่า พันได้เถิงมัคค์พละนิพพาน ก็มีด้วย

[Translation]

… of the Buddha *puṇṇapuṇṇaṃ* again and again, [What is again and again?] That is to remember by heart and recite without reading the text. Then, it should be recited as daily basis to venerate the Buddha without any missing days. This, as its fruition, would support practitioners in this lifetime and to attain the paths to Nibbāna in the future.

The similarity in the way that the term *puṇṇapuṇṇaṃ* (*punappunaṃ*) is elaborated and driven into the practical context in Lanna and Lan Xang reflects the cross-cultural transmission between the two regions. In other words, they might exchange the text and practice with one another.

## 7. Conclusions

The study of the *Dhammakāya Gāthā* has been limited in contemporary academia, so in this paper I have reviewed some of its associated scholarship in order to build up the foundational background for readers. I have presented brief information of the *Dhammakāya Gāthā*; its earliest existence in 1549 CE; textual content (the first part—personification, the second part—verses in praise, the third—summarising meditation guide); functional usages; and the locations where the *Gāthā* has been found. This section reflects the dynamic nature of the *Dhammakāya Gāthā* that circulates not only in the nations, but also "international boundaries". This contributes to the wider sphere of the study of Buddhist texts, signifying that the text is not static, but rather a "living entity" that circulates between teachers, scribes, students, rituals and other practices, depending on the decision of the users. From the textual analysis, I have challenged the interpretation of the *Dhammakāya* as a collection of the Buddha's teaching preserved in Buddhist scripture. It should rather be understood as a sublime Buddha body, contributing to awakening.

I have not only explained how the *Dhammakāya* is embodied during the Lanna Buddha image construction, but also examined Buddhist doctrines underlying the process. The reason why the image of the Buddha is understood as the living Buddha is because although the Buddha has long passed away, his *dhammakāya* still exists as if he is alive in the world. Via the construction rite, the *dhammakāya* is made visible in the bodily form so that the physical eyes of meditators can visualise and use the form as a meditation object. Once the *dhammakāya* is embodied in a Buddha image, when the *yogāvacara-s* recall the Buddha image into their mind as a mental object, responding to the call in the *Dhammakāya* text (the *dhammakāya* should be recollected by the *yogāvacara* regularly), it means that they recall or visualise the *dhammakāya*. As the result of meditation progress, they will then attain the *dhammakāya* and become Buddhas. Not only does the implantation rite tell the participants what the *dhammakāya* looks like but it also points out the location where it is present (in the belly), corresponding with the idea found in the Ayutthaya Meditation Manual. In addition to visualising a Buddha image, practitioners could recite the textual content of the *Dhammakāya Gāthā* as an alternative way to recollect the *dhammakāya*. If they understand the doctrinal meaning of the Pāli *Dhamamkāya* text, they would be able to depict the image of the *dhammakāya* in their mind.

When considering the textual meaning underlying the *Dhammakāya* text, especially its first part, "personification", with the way it is used during the mark implantation for the new Buddha image, we can see that the visual representation of the *dhammakāya* embodied in a Buddha image parallels the textual depiction of the *dhammakāya* in the *Dhamamkāya Gāthā*. The textual depiction and visual imagination of the *dhammakāya* might have been a result of a meditation experience of the *dhammakāya* (as recorded in the Ayutthaya Meditation Manual). In other words, the *Dhammakāya* text (and its visual construction) may

have been composed by someone who encountered the *dhammakāya* during the meditation practice.

　　This paper has built a foundational background for further comparative research on the Buddha image and/or Stūpa consecrations constructed in other Buddhist traditions in different locations. I hope that this research on a small collection of Buddhist texts can provide a basic methodological approach to a wider sphere of textual study in the future.

**Funding:** This paper is sponsored by Publishing Bursary Grant from the University of Otago.

**Data Availability Statement:** The literature review section in this paper has been adapted from the author's MA thesis available at http://hdl.handle.net/10523/9503 (accessed on 8 November 2023).

**Conflicts of Interest:** The author declares no conflict of interest.

## Notes

1　I use the term "Tai-Khmer" to describe a broader spectrum of Buddhist traditions, i.e., associated with cultures, practices, scripts, texts, terminologies and ideas, shared by Tai and Khmer speakers, including those from precolonial Lanna, Laos, Siam and Cambodia. The collection of these cultural schemes and practices within the cultural sphere of Tai–Khmer Buddhism is unique and cannot be found in neighbouring "Theravāda" countries, including Burma and Sri Lanka. I apply the term "traditional" [Tai–Khmer Buddhism] to signify that these cultural practices were once well known and performed by people in the area, but receive little attention today and are sometimes omitted from modern practices.

2　Today, the earliest edition of *Suttajātakanidānānisaṃsa* discovered is dated back to 1817.

3　Recently, Choompolpaisal and Skilton studied this inscription thoroughly and linked it to the *boran kammaṭṭhāna* tradition (Choompolpaisal and Skilton 2022, pp. 39–42).

4　This script was adapted from the Khmer alphabetic system in order to record a more accurate Thai pronunciation. Khom-Sukhothai script was widely used throughout central and southern Thailand during the Ayutthaya kingdom to the beginning of the twentieth century (Urkasame 2013, p. 245).

5　I would like to thank Dr. Elizabeth Guthrie for her assistance with translating this chapter from French to English.

6　*Ṭīkā Braḥ Dhammakāya* (ฎีกา/พ/ร, 1 *phuk* "fascicle") is the commentary on [the meaning of] the Body of Dhammas (henceforth abbreviated as TBD-1). It appears in a form of a palm-leaf manuscript (30 folios) written in Khom-Pāli script. This manuscript is contained within the '6322' compilation of six manuscripts that belongs to the Siamese *Tipiṭaka* collection called the *Chabap Rong Srong*, "glided palm-leaf edition". This *Tipiṭaka* edition was produced or perhaps copied sometime during 1782–1809 and sponsored by King Rāma I himself. The TBD-1 manuscript is preserved in the National Library of Thailand.

7　This manual is different from the one that Swearer studied (mentioned in the literature review).

8　In Thailand, especially northern and northeastern Thailand, when someone mentions the term "heart" (*hadaya*, ใจ), this could mean the heart as a physical organ, centre, core or navel of things such as a human, image and village. For example, in Roi-et, local peoples use the term "centre, core or navel" of the village to mean the "heart" of the village (see also https://www.silpa-mag.com/culture/article_8407 accessed on 8 November 2023).

9　This influential term has been used by scholars such as Bizot, Bernon, Crosby, Skilton, Choompolpaisal, Cholvijarn (Cholvijarn 2021) and others to describe the presence of an esoteric tradition of texts and practices, especially meditation, within the Theravāda of mainland Southeast Asia before Buddhist reformation took place during the fourth reign of the Rattanakosin dynasty of Thailand (1851–1868), i.e., that of King Mongkut. Recently, Crosby used the term "Boran Kammaṭṭhāna"—the Thai word *boran* (Khmer—*purāṇ*, and Pāli/Skt—*purāṇa*) means "traditional/ancient/old"—to identify the meditation system (written in bilingual Pāli-Khmer, bilingual Pāli-Thai and vernacular languages), which is far removed from the rationalistic monolithic Theravāda presented in many secondary sources. What makes this practice distinct from other forms of meditation is its unique approach to the body, contemplation, visualisation and soteriological paths (Crosby 2000, pp. 141–42). Coedès is the first scholar who linked the *Dhammakāya* text to *yogāvacara* tradition, as he wrote in his 1956 article that "the text, in Pāli language, entitled *Dhammakāya* or *Dhammakāyassa atthavaṇṇanā* is an opuscule belonging to the same school as the treatise published by T.W. Rhys Davids, *The Yogāvacara's Manual* (PTS, 1896) and translated by F.L. Woodward under the title Manual of a Mystic (PTS, 1916), and the *Saddavimala* described by L. Finot" (Coedès 1956, p. 254).

10　See a comparison of "Buddhānussati" among Hīnayāna, Mahāyāna and Vajrayāna in (Roe 2014) and (Harrison 1978).

11　See the traditional practice of the visualisation of the Buddha image in Burmese tradition from (Foxeus 2016). See other works on *Buddhānussti* in (Legittimo 2012; Harrison 1978).

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
