# Peer review of "Visualising the Dhammakāya through a Buddha Image: The Dhammakāya Text and Its Significance for Traditional Tai–Khmer Buddhist Practices"

_religions, doi:10.3390/rel14121446_

Round 1
Reviewer 1 Report
Comments and Suggestions for Authors
This is an important contribution to the field, well-researched and comprehensively written.
The article discusses the historical transfer and adaptation of the Pāli text, the Dhammakāya Gathā through the translocal sphere of Tai-Khmer Buddhism, focussing on paratextual instances, doctrinal changes and its adaptation to the meditative practices.
The topic discussed in the article is original and important to the study of Buddhism generally and Tai-Khmer Buddhism particularly. Methodologically, the article squarely locates itself within the new trends of studying Buddhism as transfer, translocality and circulation elaborated, for example, in the works of Neelis (2011). Besides this, the greatest originality of this article lies in the close textual study of the Gathā in connection with the material dimension of image-making.
I think the article adds to the similar publications in Buddhist studies because of its interdisciplinary approach to the presented material, combining with itself textual studies, manuscript studies, material culture and specific practices (ritual studies) relevant to contemporary Buddhist practice.
I think that the methodological riches is the article's greatest strength. If improvements need to be implemented, I would suggest adding a few names of scholars who study Buddhism through the lens of transfer and circulation, such as the already mentioned Neelis or Walters (1999) who discusses the agency of Southeast Asian people through the concept of "localization".
Neelis, Jason. 2011. Early Buddhist Transmission and Trade Networks: Mobility and Exchange within and beyond the Northwestern Borderlands of South Asia. Leiden/Boston: Brill.
Wolters, Oliver W. 1999 [1982]. History, Culture and Region in Southeast Asian Perspectives. Singapore:Institute of Southeast Asian Studies.
Some references to the Buddhist visualization practices could be included, for example, an important study by Andy Rotman, "Thus I Have Seen: Visualizing Faith in Early Buddhism" The conclusions are clearly stated and conclude very well the main arguments developed throughout the article.
Author Response
Thank you very much for taking the time to review this manuscript. Please find the detailed responses below and the corresponding revisions/corrections highlighted in the re-submitted files.
“I think that the methodological riches is the article's greatest strength. If improvements need to be implemented, I would suggest adding a few names of scholars who study Buddhism through the lens of transfer and circulation, such as the already mentioned Neelis or Walters (1999) who discusses the agency of Southeast Asian people through the concept of "localization".
Neelis, Jason. 2011. Early Buddhist Transmission and Trade Networks: Mobility and Exchange within and beyond the Northwestern Borderlands of South Asia. Leiden/Boston: Brill.
Wolters, Oliver W. 1999 [1982]. History, Culture and Region in Southeast Asian Perspectives. Singapore:Institute of Southeast Asian Studies (I included it in the reference).
Some references to the Buddhist visualization practices could be included, for example, an important study by Andy Rotman, "Thus I Have Seen: Visualizing Faith in Early Buddhism" (I discussed it and included it in the reference). The conclusions are clearly stated and conclude very well the main arguments developed throughout the article.”
Best Regards,
Woramat
Reviewer 2 Report
Comments and Suggestions for Authors
There is some very interesting data in this article and it would be great to bring it together as an article.
I think the author needs the support of their supervisor or mentor to sort out what they are trying to do the article, however. There are unsubstantiated arguments, the use of personal belief as a source of argument, and a lack of clarity of argument and cross-referencing. The overall purpose of the article is unclear. A couple of claims to be overturning previous scholarship do not seem justified.
I am sure there is an important article to be made of this material, but somehow it feels as if the thinking has not been clarified before putting pen to paper. It reads as rather an amorphous lump of information. As a reader I want clear evidence and clear arguments.
I therefore recommend publication only after major revision. I have provided detailed comments throughout. I have mostly not put pages but they are in order.
Daniel Stuart’s discussion of the term dharmakāya needs to be included.
The literature review section 2 should surely include Choompolpaisal and Skilton’s recent article on the same (which is in the bibliography)? I think it would also be good to have a sentence or half sentence to complete the understanding of what Trent Walker related it to, the reference to which is currently too brief to understand.
Below I go through the text in order, so English language corrections are interlineated with more substantive points.
p.1 line 14 Little attention by > from
the most significant > its most significant or a significant
Rather than “see in”, us “for example” or “as in” and give us the reference to the location. Also need specific reference for Khuddakanikāya.
“have received little attention by the modern practitioners” This phrasing is used for scholarship, not for practitioners. You could say that it is little used in modern practice, except for…(image consecration?).
“an engraved stone slab” > “the engraved stone slab mentioned above, …” Check in this entire section that you have appropriate cross-referencing to earlier.
“caturiddhipādañāṇa” remember that hopefully not all readers of your article will be Theravada specialists and explain terms such as these and their context if you want a broader readership. The same goes for the image. It would be a very useful image if the equated items were also given in translation.
p.5 It is not clear what the second part consists of.
The translation is incomplete – sabbannuta missing, an>a: “This Buddha’s characteristic is the dhammakāya. [This] should 187 be remembered again and again by a noble man of a good family, practicing 188 spiritual exercises, possessing sharp Knowledge and aspiring to become an 189 Buddha...”
The reference for the quotations from Jones has been included in the quotation markers. I would be inclined to contextualise the quotation because of the Mahayana uses of the term. It would be good to include the discussion of the term in Daniel Stuart’s Saddharmanusmṛti book.
“One might be curious as to how the dhammakāya is constructed, since it was under- 200 stood by some scholars as a formless body, invisible body, and a collection of the Buddha 201 teachings preserved in the Tipiṭaka. In this section, I shall present the most important pro- 202 cedure when a Buddha image is constructed in Lanna and suggest how the concept of the 203 dhammakāya is understood by Buddhists in the region.” In this paragraph, you need to acknowledge context.
“The manuscripts write” > “It is stated in the manuscripts” (which manuscripts, could you not just refer us to the text)
The Manual for Installing. Is this different or separate from the work used by Swearer. Make clear and link back to lit review.
“One of the best examples of a paratext” How many such paratexts are there and what makes this one of the best?
Provide evidence for this statement: “The first interesting point to note in this passage is that the term “yogāvacara-kulaputta” here is not restricted to the son of a good family but includes all spiritual practitioners, both men and women.”
The statement “The location where the golden casket is placed also suggests the point where the dhammakāya is located (in the navel or the centre of the body)” does not make sense unless one is now using one of the other meanings of dhammakaya or slipping into thinking of Dhammakaya meditation. Also need to explain what Ayutthaya Meditation Manual is.
In Figure 2 labels, it is surely misleading to state verses “such as”. Surely it is just those verses and not others like those verses. The statement “caskets used to be placed in the belly of the image” is unclear. Are these the contents including caskets of the kind once place in the belly of the image via the back? It might be an idea to explain the other items in the picture. It looks as if there are physical body relics too?
I don’t understand the logic of or evidence for this argument. Also, a source for the quotation is needed: “Perhaps, after knowing where the dhammakāya resides, experienced teachers also wished to demonstrate what the dhammakāya looked like to Buddhist students. Therefore, they installed the elements of the dhammakāya into the marks of the Buddha image, demonstrating that the dhammakāya is a Buddha likeness as written in the text: dhammakāya-buddhalakkhaṇaṃ “this set of Buddha marks is [called] the dhammakāya.” Please also check the Swearer quote word by word. The grammar is a little odd. The Vakkalisutta statement is a bit different, since it equates the Buddha with the dhammakaya and not the imagem so I think a step in your thinking needs to be made explicit here.
“mediation practitioners” Meditation-mediation is such a common typo, it’s worth doing a search in any article one writes about meditation.
“Sabbaññūbuddha “a buddha who awakens by himself”” This is not an accurate translation of Sabbaññūbuddha.
Figure 3. I would think the things shown in this image would be worth pointing out. What is very striking is that these elements are all components or knowledges that the Buddha realised on awakening. So the image is being awakened by their insertion. Rather than “a pair of noble breasts” > “the pair…” (I assume the Buddha only has one set.)
“they linked yogāvacara” > they link yogāvacara (use present tense referring to contents of scholarship)
“a good sone from a good family” is this an overtranslation of kulaputta?
“understood as the body opposite to the physical body of the Buddha” rephrase, not “opposite”
Odd use of the Mahayana phrase “skilful means” in this context and without explanation.
“The text writes that:” > “The text reads”
You need to sort out the translation on p.11. There is not explanation given of what is in brackets, some of which is unnecessary for meeting, what is underlined, what is in bold, and there’s quite a long passage you have added in brackets at the end. I would also expect to see something written after the translation so the reader knows how you are trying to use this material.
The cricism that previous authors don’t give enough reason for identifying the text as a meditation text seems odd given that the text itself, as you yourself quote, instructs its use for meditation.
“kong” use “kruba kong” i.e. the form of the name you gave previously.
“would be able to visualise or recall the image” Watch out – whenever a scholar writes “would …” in this way, it means they are speculating without evidence. It’s very common in art history. See if you can write with more precision or clearly identify that this is entirely your imagination.
“The same recommendation appears..” redo sentence. It is ungrammatical, and also rather unclear. Divide into two? Also with the texts, remind us which you have mentioned before and which are new introductions.
can be varied > can vary
“locating the text to a practical orientation.” Rephrase. Not idiomatic.
“I have challenged the interpretation 449 of the Dhammakāya as a collection of the Buddha preserved in Buddhist scripture” Rephrase for two reasons: 1) it is not grammatical; 2) the meaning is unclear, but if you are claiming that you are the first to associate dhammakāya with the qualities of the Buddha rather than the collection of the qualities of the Buddha, then your claim is untrue. Stuart (mentioned above) does a decent survey of the discussion of this multivalent phrase.
“a sublime Buddha likeness body that consists of head, nose, 451 hands, ears etc. “ Again, rephrase. It is unclear, in part because you can’t just form list-compounds in English.
“construction procedure on how to construct” sort out the grammar here
p.14 You need to sort out the issue of the qualities being associated with the different parts of the body and being at the navel. You haven’t yet explained what is meant by this.
“parallels with” > “parallels” (no with)
“The textual depiction and visual imagination of the dhammakāya might 474 have been a result of a meditation experience of the dhammakāya (as recorded in the Ayutthaya Meditation Manual). In other words, the Dhammakāya text (and its visual construction) may have been composed by someone who encounters the dhammakāya during the meditation practice.” This statement is unsupportable, so just state it as your personal belief if you want to include it. It is not supported by Anālayo’s statement, which is about the connection between visual art and texts, and meditation. He is not saying that the meditation is the original inspiration for either, as you seem to be saying
Comments on the Quality of English Language
There are slight infelicities of English throughout that would be easier to address using track changes if the one received a word doc. Also too many spaces, some doubling up of phrasing, some fonts odd. I have gone through in my other comments with suggestions for many of the issues.
Author Response
Thank you very much for taking the time to review this manuscript. Please find the detailed responses below and the corresponding revisions/corrections yellow highlighted in the re-submitted files.
Daniel Stuart’s discussion of the term dharmakāya needs to be included. (I cannot find the name of the book or article, can you please give me the full name?)
The literature review section 2 should surely include Choompolpaisal and Skilton’s recent article on the same (which is in the bibliography)? I think it would also be good to have a sentence or half sentence to complete the understanding of what Trent Walker related it to, the reference to which is currently too brief to understand. (I added sentences from Trent’s analysis of the Dhammakāya Gāthā from his thesis and included Choompolpaisal and Skilton’s recent article in the footnote under the work of Thongkhamwan as they studied the same inscription.)
Below I go through the text in order, so English language corrections are interlineated with more substantive points.
p.1 line 14 Little attention by > from (corrected)
the most significant > its most significant or a significant (I omitted this phrase.)
Rather than “see in”, us “for example” or “as in” and give us the reference to the location. Also need specific reference for Khuddakanikāya. (I corrected and provided references for Khuddahanikāya and a scholar who studied this matter.)
“have received little attention by the modern practitioners” This phrasing is used for scholarship, not for practitioners. You could say that it is little used in modern practice, except for…(image consecration?). (I have revised the sentence.)
“an engraved stone slab” > “the engraved stone slab mentioned above, …” Check in this entire section that you have appropriate cross-referencing to earlier. (I corrected the section 3 for cross-referencing.)
“caturiddhipādañāṇa” remember that hopefully not all readers of your article will be Theravada specialists and explain terms such as these and their context if you want a broader readership. The same goes for the image. It would be a very useful image if the equated items were also given in translation. (I mentioned references. If someone would be interested in the full details of the Dhammakāya Gāthā, they can look at the full translation from the given references.)
p.5 It is not clear what the second part consists of. (corrected-I have added a translation of the second part in the section)
The translation is incomplete – sabbannuta missing, an>a: “This Buddha’s characteristic is the dhammakāya. [This] should 187 be remembered again and again by a noble man of a good family, practicing 188 spiritual exercises, possessing sharp Knowledge and aspiring to become an 189 Buddha...” (corrected by putting an “omniscient” buddha…)
The reference for the quotations from Jones has been included in the quotation markers. I would be inclined to contextualise the quotation because of the Mahayana uses of the term. It would be good to include the discussion of the term in Daniel Stuart’s Saddharmanusmṛti book. (I cannot access this book at my university and have to wait a few weeks for interloan if needed the book. I am not so sure if I can respond to your suggestion here because I am given only 10 days to revise the article.)
“One might be curious as to how the dhammakāya is constructed, since it was under- 200 stood by some scholars as a formless body, invisible body, and a collection of the Buddha 201 teachings preserved in the Tipiṭaka. In this section, I shall present the most important pro- 202 cedure when a Buddha image is constructed in Lanna and suggest how the concept of the 203 dhammakāya is understood by Buddhists in the region.” In this paragraph, you need to acknowledge context. (revised-“In this section, I shall present the way that the dhammakāya is embodied during a Buddha image construction in Lanna (see figure 3), reflecting how the concept of the dhammakāya is understood by Buddhists in the region.”)
“The manuscripts write” > “It is stated in the manuscripts” (which manuscripts, could you not just refer us to the text) (corrected)
The Manual for Installing. Is this different or separate from the work used by Swearer. Make clear and link back to lit review. (put the clarification in the footnote)
“One of the best examples of a paratext” How many such paratexts are there and what makes this one of the best? (revised this sentence to “a good example of the paratext”…)
Provide evidence for this statement: “The first interesting point to note in this passage is that the term “yogāvacara-kulaputta” here is not restricted to the son of a good family but includes all spiritual practitioners, both men and women.” (I highlighted the words in bold in the translation and added a sentence “according to the above quotation.”)
The statement “The location where the golden casket is placed also suggests the point where the dhammakāya is located (in the navel or the centre of the body)” does not make sense unless one is now using one of the other meanings of dhammakaya or slipping into thinking of Dhammakaya meditation. Also need to explain what Ayutthaya Meditation Manual is. (I revised the sentence and explained that the term “heart or hadaya” could mean “core, centre, or navel” of things (human, village or images). I also added the reference where the reader could read on historical background and details of the manual.)
In Figure 2 labels, it is surely misleading to state verses “such as”. Surely it is just those verses and not others like those verses. The statement “caskets used to be placed in the belly of the image” is unclear. Are these the contents including caskets of the kind once place in the belly of the image via the back? It might be an idea to explain the other items in the picture. It looks as if there are physical body relics too? (I revised the sentence into “A golden casket containing the three Gāthā-s was placed in the belly of the image. Other valuable objects such as crystal balls, the heart of the Buddha made from silver and physical relics were also installed inside the image.”)
I don’t understand the logic of or evidence for this argument. Also, a source for the quotation is needed: “Perhaps, after knowing where the dhammakāya resides, experienced teachers also wished to demonstrate what the dhammakāya looked like to Buddhist students. Therefore, they installed the elements of the dhammakāya into the marks of the Buddha image, demonstrating that the dhammakāya is a Buddha likeness as written in the text: dhammakāya-buddhalakkhaṇaṃ “this set of Buddha marks is [called] the dhammakāya.” Please also check the Swearer quote word by word. The grammar is a little odd. The Vakkalisutta statement is a bit different, since it equates the Buddha with the dhammakaya and not the imagem so I think a step in your thinking needs to be made explicit here. (I revised to “The above quotation suggests that the dhammakāya is attainable and could be visioned during the insight (Vipassanā) meditative state. Perhaps, after encountering the dhammakāya during the meditation, the author of the Dhammakāyatext wrote that dhammakāya-buddhalakkhaṇaṃ “this set of Buddha marks is [called] the dhammakāya,” instructing what thedhammakāya looks like. When this idea was transmitted in the ritual context, the set of the dhammakāya’s marks is installed into the marks of the Buddha image, demonstrating that virtually the dhammakāya is a buddha likeness.”)
“mediation practitioners” Meditation-mediation is such a common typo, it’s worth doing a search in any article one writes about meditation. (corrected)
“Sabbaññūbuddha “a buddha who awakens by himself”” This is not an accurate translation of Sabbaññūbuddha. (deleted this sentence)
Figure 3. I would think the things shown in this image would be worth pointing out. What is very striking is that these elements are all components or knowledges that the Buddha realised on awakening. So the image is being awakened by their insertion. Rather than “a pair of noble breasts” > “the pair…” (I assume the Buddha only has one set.) (I put “see figure 3” in the 2nd paragraph of section 4 because it relates with textual components of the Dhammakāya Gāthā and I put an explanation in the section.)
“they linked yogāvacara” > they link yogāvacara (use present tense referring to contents of scholarship) (corrected)
“a good sone from a good family” is this an overtranslation of kulaputta? (reduced to “a son from a good family”)
“understood as the body opposite to the physical body of the Buddha” rephrase, not “opposite” (…the body, in contrast to the physical body…)
Odd use of the Mahayana phrase “skilful means” in this context and without explanation. (changed to “a Dhamma puzzle”)
“The text writes that:” > “The text reads” (corrected)
You need to sort out the translation on p.11. There is not explanation given of what is in brackets, some of which is unnecessary for meeting, what is underlined, what is in bold, and there’s quite a long passage you have added in brackets at the end. I would also expect to see something written after the translation so the reader knows how you are trying to use this material. (I removed the bold from some words and wrote an explanation below the quotation.)
The criticism that previous authors don’t give enough reason for identifying the text as a meditation text seems odd given that the text itself, as you yourself quote, instructs its use for meditation. (what I meant was to a particular meditation practice “Yogāvacara/Boran Kammaṭṭhāna.”)
“kong” use “kruba kong” i.e. the form of the name you gave previously. (corrected)
“would be able to visualise or recall the image” Watch out – whenever a scholar writes “would …” in this way, it means they are speculating without evidence. It’s very common in art history. See if you can write with more precision or clearly identify that this is entirely your imagination. (I put “I speculate.”)
“The same recommendation appears..” redo sentence. It is ungrammatical, and also rather unclear. Divide into two? Also with the texts, remind us which you have mentioned before and which are new introductions. (revised the sentence to “the similar practice can be found in Central Thailand and Lan Xang…”)
can be varied > can vary (corrected)
“locating the text to a practical orientation.” Rephrase. Not idiomatic. (revised it into two sentences-- It suggests how the Dhammakāya text is used in a practical orientation.)
“I have challenged the interpretation 449 of the Dhammakāya as a collection of the Buddha preserved in Buddhist scripture” Rephrase for two reasons: 1) it is not grammatical; 2) the meaning is unclear, but if you are claiming that you are the first to associate dhammakāya with the qualities of the Buddha rather than the collection of the qualities of the Buddha, then your claim is untrue. Stuart (mentioned above) does a decent survey of the discussion of this multivalent phrase. (revised to “a collection of the Buddha’s Teaching”…)
“a sublime Buddha likeness body that consists of head, nose, 451 hands, ears etc. “ Again, rephrase. It is unclear, in part because you can’t just form list-compounds in English. (revised to a sublime Buddha body, contributing to awakening)
“construction procedure on how to construct” sort out the grammar here (revised the sentence to “I have not only explained how the Dhammakāya is embodied during the Lanna Buddha image construction,”…)
p.14 You need to sort out the issue of the qualities being associated with the different parts of the body and being at the navel. You haven’t yet explained what is meant by this. (I explained in the 7th paragraph of section 4)
“parallels with” > “parallels” (no with) (corrected)
“The textual depiction and visual imagination of the dhammakāya might 474 have been a result of a meditation experience of the dhammakāya (as recorded in the Ayutthaya Meditation Manual). In other words, the Dhammakāya text (and its visual construction) may have been composed by someone who encounters the dhammakāya during the meditation practice.” This statement is unsupportable, so just state it as your personal belief if you want to include it. It is not supported by Anālayo’s statement, which is about the connection between visual art and texts, and meditation. He is not saying that the meditation is the original inspiration for either, as you seem to be saying (deleted Analayo’s statement)
Best regards
Reviewer 3 Report
Comments and Suggestions for Authors
This excellent article makes an important contribution to our understanding of the term Dhammakāya, and its ritual and devotional translation into a physical ‘body.’ Closely argued and carefully analytic, it introduces the Dhammakāya Gāthā for consideration: the argument shows how study of this text illuminates the known corpus of material on this subject. The strong recommendation is for it to be pubished.
One small point:
composed not composted (l 113 page 3)
Author Response
Thank you very much for taking the time to review this manuscript. Please find the detailed responses below and the corresponding revisions/corrections in yellow highlighted in the re-submitted files.
"One small point:
composed not composted (l 113 page 3)
I corrected.
Best Regards